# TabPFN-TSRA: Retrieval-Augmented TabPFN for Time-Series Forecasting

Zijian Tang [1]   Ying Zhang [1]   Zhenjun Liu [2]   Sibo Cai [3]   Shibin Yue [2]   Meili Zhang [2]

## Abstract

TabPFN-based time-series forecasting enables zero-shot prediction by reformulating forecasting as tabular regression, but existing methods still rely on explicit temporal features and global historical summarization, which can be insufficient for non-stationary, event-driven dynamics and may dilute informative local signals. We propose TabPFN-TSRA, a retrieval-augmented framework for TabPFN-based forecasting. TabPFN-TSRA retrieves analogous historical patterns using level-shift-corrected correlation matching and organizes the retrieved segments into temporally aligned feature matrices for TabPFN. This design provides compact and relevant local evidence beyond fixed features while preserving temporal structure. Experiments on 11 benchmark datasets show that TabPFN-TSRA outperforms TabPFN-TS and TabPFN-TSP, achieving an 82% win rate in terms of MAE and enhancing the ability to forecast trend evolution. These results demonstrate the effectiveness of retrieval augmentation for TabPFN-based forecasting. Code is available at: https://github.com/sibo-cai/TabPFN-TSRA.

## 1. Introduction

Time-series forecasting is a fundamental task across a wide range of real-world applications, where accurate forecasts are crucial for planning, monitoring, and decision-making (Benevenuto et al., 2009; Martín et al., 2010; Granger & Newbold, 2014). Recent advances in TabPFN-TS demonstrate that forecasting can be formulated as tabular regression, enabling zero-shot prediction through the in-context learning capability of TabPFN (Hoo et al., 2024). This lightweight and training-free paradigm offers a promising direction toward general-purpose forecasting across diverse time series.

Despite its promise, TabPFN-TS and its extension TabPFN-TSP still encode temporal structure mainly through explicit features, including calendar variables and FFT-estimated frequencies for periodic patterns (Cai et al., 2025). While these features are useful for regular cycles, real-world time series often exhibit non-stationary and event-driven dynamics, with patterns varying in period, shape, and recurrence (Kim et al., 2021). For instance, store demand may gradually rise before a promotion and drop sharply afterward, forming a sparse local pattern that fixed frequency features can hardly capture. Moreover, TabPFN-TS summarizes the full historical context in a global manner, which can dilute informative local signals and reduce its sensitivity to prevailing temporal trends (Potapczynski et al., 2026).

Retrieval augmentation has been widely adopted in NLP to provide relevant context through the input, and has also shown promise in time-series forecasting (Han et al., 2025; Li, 2025). Inspired by this paradigm, we propose **TabPFN-TSRA**, a retrieval-augmented framework for TabPFN-based time-series forecasting. The core idea is to retrieve analogous patterns from historical sequences and transform the retrieved segments into a temporally aligned feature matrix for TabPFN. This design helps capture complex local dynamics beyond fixed temporal features and alleviates the dilution of informative signals in long contexts, thereby improving forecasting performance.

We evaluate TabPFN-TSRA on 11 time-series benchmark datasets. The results show that TabPFN-TSRA achieves an 82% win rate over TabPFN-TS and its variant TabPFN-TSP in terms of MAE. Moreover, TabPFN-TSRA helps alleviate the difficulty of TabPFN-TS in forecasting sustained trends, especially when future values extend beyond the observed target range. These results demonstrate the effectiveness of retrieval augmentation for TabPFN-based forecasting.

## 2. Methodology

### 2.1. Overview

**Problem Definition.** Given a univariate time series $L = [x_1, x_2, \ldots, x_N]$ and a forecasting horizon $m$, we use the observed history $R = [x_1, \ldots, x_{N-m}]$ to predict the terminal target sequence $y_{\text{test}} = [x_{N-m+1}, \ldots, x_N]$, without observing any future values.

[1]Peking University [2]Guizhou Xijiu [3]The Open University of China. Correspondence to: Sibo Cai <caisb@ouchn.edu.cn>.

*Proceedings of the $2^{nd}$ ICML Workshop on Foundation Models for Structured Data*, Seoul, South Korea. 2026. Copyright 2026 by the author(s).

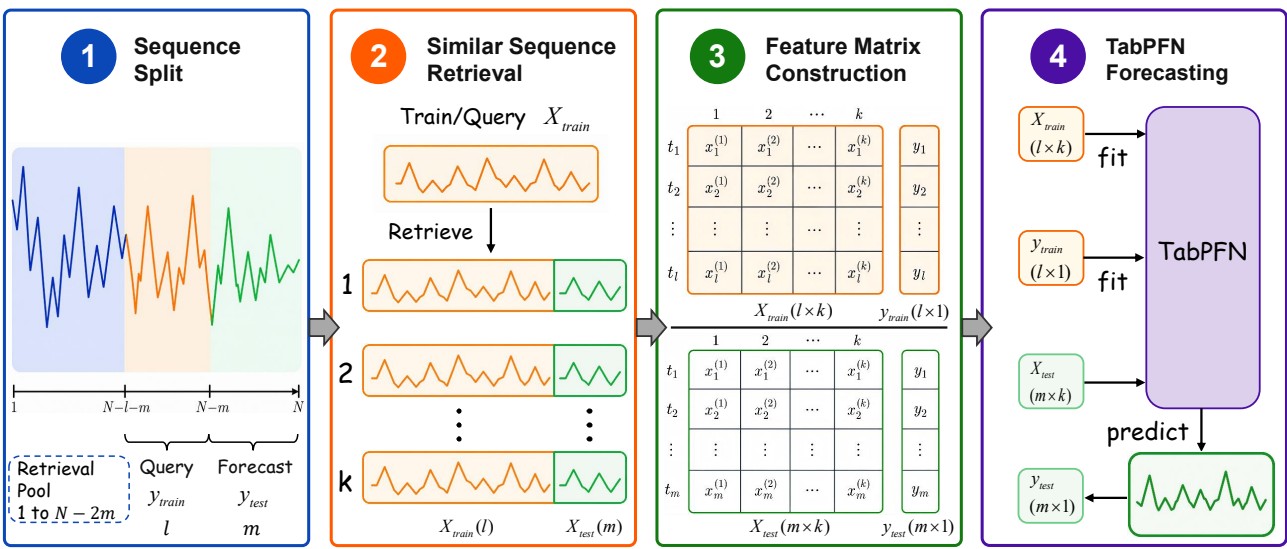

*Figure 1.* The overall framework of our TabPFN-TSRA.

We propose **TabPFN-TSRA**, a retrieval-augmented framework for TabPFN-based time-series forecasting. The core idea is to retrieve analogous patterns from historical observations and transform the retrieved segments into temporally aligned feature matrices for TabPFN, enabling the model to capture local dynamics beyond fixed temporal features and reducing the dilution of informative signals in long contexts. The proposed method consists of four main steps. First, the observed history is divided into a retrieval pool and a query segment used for similarity search. Second, we retrieve from the pool a set of subsequences that are most similar to the query segment, together with their subsequent trajectories. Third, the retrieved subsequences are temporally aligned to construct the input feature matrices for TabPFN. Finally, the constructed matrices are fed into TabPFN to generate the forecast. The overall framework is illustrated in Figure 1.

### 2.2. Retrieval Module Architecture

**Train-Query Split.** Given the observed history $R = [x_1, \ldots, x_{N-m}]$, we reserve its last $l$ observations as the query segment, denoted by $\mathbf{q} = [x_{N-m-l+1}, \ldots, x_{N-m}]$. This segment immediately precedes the forecasting horizon and serves as the reference pattern for identifying analogous historical contexts. Since each retrieved subsequence must be paired with its subsequent trajectory of length $m$, the retrieval pool is restricted to the earlier prefix $[x_1, \ldots, x_{N-2m}]$, ensuring that the continuation following any retrieved subsequence is fully observed within $R$ and does not overlap with the unseen target sequence $y_{\text{test}}$.

**Level-Shift Correction.** When comparing the query segment with candidates from the retrieval pool, their absolute

levels may differ even if they exhibit similar temporal dynamics. To focus the comparison on relative shape rather than absolute magnitude, we apply the same level-shift correction to both the query and each candidate subsequence before computing their similarity. Specifically, for any length-$l$ subsequence $\mathbf{s} = [s_1, \ldots, s_l]$, we take its last observation $s_l$ as the local reference value and subtract it from every time step:

$$\tilde{s}_i = s_i - s_l, \quad i = 1, \ldots, l, \tag{1}$$

Thus, the corrected query and candidate subsequences are compared after being aligned, which reduces the influence of local level differences while preserving their relative temporal patterns (Zeng et al., 2023).

**Correlation-Guided Analogue Retrieval.** After level-shift correction, we retrieve historical analogues whose relative variations are most consistent with the query segment. To reduce the influence of scale changes and residual value offsets, we use the Pearson correlation coefficient as the similarity measure. For each candidate subsequence $\mathbf{u}_s = [x_s, \ldots, x_{s+l-1}]$ from the retrieval pool, we compare its corrected form $\tilde{\mathbf{u}}_s$ with the corrected query $\tilde{\mathbf{q}}$. We retain only candidates whose correlation exceeds a minimum similarity threshold $\tau$:

$$\mathcal{C}_\tau = \left\{ s \in \{1, \ldots, N-2m-l+1\} \mid \rho(\tilde{\mathbf{q}}, \tilde{\mathbf{u}}_s) \geq \tau \right\}, \tag{2}$$

Among the retained candidates, we select at most $k$ subsequences with the highest correlations. Let $r = \min(k, |\mathcal{C}_\tau|)$. The selected analogue set is defined as

$$\mathcal{S}_k = \underset{\substack{\mathcal{S} \subseteq \mathcal{C}_\tau \\ |\mathcal{S}| = r}}{\arg\max} \sum_{s \in \mathcal{S}} \rho(\tilde{\mathbf{q}}, \tilde{\mathbf{u}}_s). \tag{3}$$

### 2.3. Feature Matrix Construction and Prediction

**Construction of $y_{\text{train}}$.** We define the train target as the query segment. Specifically, $y_{\text{train}}$ is set to the length-$l$ subsequence immediately preceding the forecasting horizon. The forecasting target $y_{\text{test}}$ is the subsequent length-$m$ sequence to be predicted. In this way, $y_{\text{train}}$ captures the latest observed temporal dynamics, while $y_{\text{test}}$ remains fully unobserved during model input construction.

**Construction of $X_{\text{train}}$ and $X_{\text{test}}$.** We then transform the retrieved historical analogues into feature matrices for TabPFN. For each retrieved candidate, we keep both the matched length-$l$ subsequence and its subsequent length-$m$ trajectory. The matched subsequences are aligned with $y_{\text{train}}$ according to their relative temporal indices and stacked column-wise to form $X_{\text{train}}$, while their subsequent trajectories are aligned with the forecasting horizon and stacked column-wise to form $X_{\text{test}}$. To preserve the original magnitude information of the time series, the values used in $X_{\text{train}}$, $X_{\text{test}}$, and $y_{\text{train}}$ are taken from the raw sequence directly, without applying level-shift correction used only for similarity computation.. During matrix construction, rows are aligned by the relative time indices within each analogue, so values in the same row correspond to the same relative time step across retrieved analogues. This temporal-index alignment preserves the intra-analogue temporal dynamics, such as trend evolution, providing TabPFN with a structured representation for modeling aligned historical patterns.

**Prediction with TabPFN.** Finally, we feed the constructed tabular context into TabPFN for forecasting. The pair $X_{\text{train}}$ and $y_{\text{train}}$ serves as the in-context regression examples, allowing TabPFN to learn how the retrieved analogues relate to the recent observed target values, while $X_{\text{test}}$ is used as the query input to forecast the future horizon. The resulting output is taken as the forecast sequence.

## 3. Experiments

### 3.1. Experimental Setup

**Datasets and Evaluation Metrics.** We evaluate TabPFN-TSRA on 11 benchmarks with diverse domains, dataset lengths, and frequencies. The datasets include ETT-small with four electricity transformer subsets (ETTh1, ETTh2, ETTm1, and ETTm2), Electricity, Exchange Rate, Weather, Traffic, Illness following (Wu et al., 2021; Zhou et al., 2021; Wu et al., 2022), AirPassengers (Box et al., 2015), and Stock collected via AkShare (King & Zhang, 2022). To ensure a unified setting, we convert all datasets into univariate forecasting tasks by retaining only the target series for multivariate datasets, using the `OT` column when available. For Stock, we use the `close` column as the target, as it reflects the final market valuation within each trading period

and is commonly used for stock prediction.

Since our method does not involve additional normalization, all series are kept in their original scales. For most datasets, we follow previous forecasting settings and use forecasting horizons $F \in \{96, 192, 336, 720\}$. For Illness, we use $F \in \{24, 36, 48, 60\}$ following prior work (Nie et al., 2022; Wang et al., 2024). For Stock and AirPassengers, we adjust the forecasting horizons according to their dataset lengths to ensure feasible evaluation settings, using $F \in \{24, 48, 60, 96\}$ for Stock and $F \in \{1, 3, 6, 12\}$ for AirPassengers. We consider two metrics for evaluation: mean squared error (MSE) and mean absolute error (MAE).

**Baselines.** We compare TabPFN-TSRA with two representative TabPFN-based time-series forecasting methods: TabPFN-TS and TabPFN-TSP. These two baselines reflect existing ways of adapting TabPFN to forecasting tasks. By comparing against them, we evaluate whether retrieval augmentation can further improve TabPFN-based forecasting.

**Implementation Details.** For TabPFN-TSRA, we set the number of retrieved subsequences to $k = 50$ and the minimum similarity threshold to $\tau = 0.8$ for most datasets. For Stock and AirPassengers, due to their relatively limited sequence lengths, we set $k = 5$ to avoid introducing excessive or weakly related retrieval candidates. The train length $l$ is set according to dataset-specific temporal characteristics. Further implementation details are provided in Appendix C. Additional regressor replacement study, hyperparameter study, and qualitative analyses are presented in Appendix B.

### 3.2. Experimental Results

#### 3.2.1. MAIN RESULTS

Table 1 summarizes the overall performance. The results reveal several key findings: **First**, TabPFN-TSRA outperforms the two TabPFN-based baselines, achieving a win rate of 82% in terms of MAE, demonstrating the effectiveness of our retrieval augmentation. **Second**, on trend-dominated datasets such as Illness, TabPFN-TSRA reduces MAE and MSE by about 56% and 79% compared with TabPFN-TS, suggesting that the constructed feature matrices better leverage TabPFN to predict trend changes. **Third**, on complex non-stationary datasets such as AirPassengers, TabPFN-TSRA achieves substantially lower errors than TabPFN-TSP, suggesting that retrieved local evidence provides more effective forecasting cues than explicit frequency-based temporal features. **Finally**, TabPFN-TSRA achieves strong performance with more compact feature matrices. Compared with TabPFN-TS, which typically uses around $4096 \times 28$ input matrices, TabPFN-TSRA uses matrices of $(l + m) \times k$. This retrieval-augmented design avoids relying on the entire long historical context and instead selects the most relevant

*Table 1.* Main forecasting results on 11 benchmark datasets. We report the average results over all forecasting horizons. Lower MSE and MAE indicate better performance. The best results are highlighted in bold. Scientific notation with e-notation is used for large values.

| Dataset | TabPFN-TSRA | | TabPFN-TSP | | TabPFN-TS | |
|---|---|---|---|---|---|---|
| | MAE | MSE | MAE | MSE | MAE | MSE |
| Traffic | **0.0025** | **0.0000** | 0.0029 | **0.0000** | 0.0032 | **0.0000** |
| Exchange | **0.0427** | **0.0021** | 0.0428 | 0.0027 | 0.0503 | 0.0046 |
| Electricity | 203.07 | 8.01e4 | 184.05 | 6.56e4 | **181.02** | **6.44e4** |
| ETTh1 | **2.42** | **9.83** | 2.55 | 10.41 | 3.21 | 15.76 |
| ETTh2 | **4.98** | **41.86** | 5.66 | 51.10 | 5.04 | 44.34 |
| ETTm1 | **1.69** | 5.48 | 2.18 | 8.12 | 1.72 | **5.28** |
| ETTm2 | **3.54** | **25.17** | 5.49 | 48.66 | 3.81 | 26.51 |
| Stock | **1.618** | **5.366** | 2.102 | 7.495 | 1.805 | 6.130 |
| Illness | **1.39e5** | **3.83e10** | 2.75e5 | 9.67e10 | 3.14e5 | 1.79e11 |
| AirPassengers | **13.63** | **301.67** | 65.53 | 8089.78 | 15.45 | 360.79 |
| Weather | 10.48 | 197.07 | 9.81 | **164.13** | **9.37** | 166.64 |

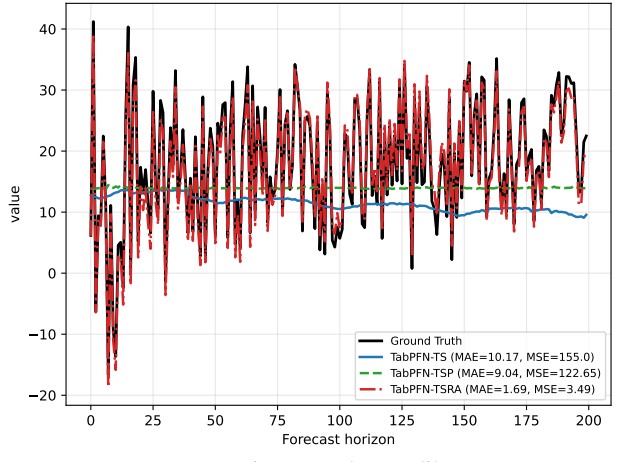

(a) Rare non-stationary and event-like pattern

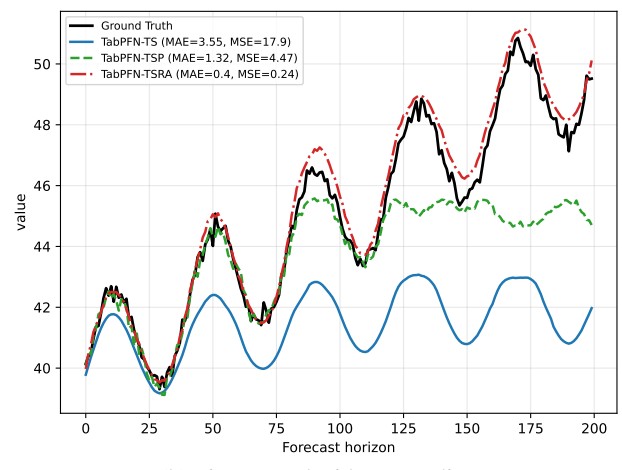

(b) Linear trend with seasonality

*Figure 2.* Qualitative results on two synthetic scenarios generated following RAFT (Han et al., 2025) and TabPFN-TS (800 history points and a 200-point forecasting horizon). Scenario (a) contains a non-stationary, event-like pattern that appears only rarely in the history.

historical subsequences, thereby concentrating informative local signals and alleviating signal dilution.

### 3.2.2. QUALITATIVE ANALYSIS

Figure 2 shows the qualitative results. **First**, in the rare non-stationary and event-like scenario, TabPFN-TSRA can retrieve rare but relevant historical analogues to guide forecasting, whereas TabPFN-TSP is limited by explicitly designed periodic features and TabPFN-TS may dilute such sparse informative signals in the global context. **Second**, in the linear-trend-with-seasonality scenario: (1) compared with TabPFN-TS, TabPFN-TSRA follows the underlying trend more closely, suggesting that the temporally aligned feature matrices, which preserve temporal order within retrieved analogues, provide a more suitable structure for trend forecasting while retaining the ability to capture periodic patterns without explicit temporal features; (2) compared with TabPFN-TSP, TabPFN-TSRA is more robust to level shifts

and noise perturbations. Under such disturbances, TabPFN-TSP matches only one historical reference sequence, which can make it difficult to capture evolving trends and more sensitive to noise. In contrast, TabPFN-TSRA retrieves multiple reference sequences through level-shift-corrected correlation matching, providing richer evidence for extrapolating dynamics.

## 4. Conclusion

We propose **TabPFN-TSRA**, a retrieval-augmented framework for TabPFN-based time-series forecasting. It retrieves analogous historical patterns and organizes them into temporally aligned feature matrices, providing compact local evidence beyond fixed temporal features and global historical context. Experiments show that TabPFN-TSRA outperforms TabPFN-TS and TabPFN-TSP, validating retrieval augmentation for TabPFN-based forecasting.

## Acknowledgments

This work is supported by the scientific research project of Engineering Research Center of Integration and Application of Digital Learning Technology, Ministry of Education (Grant No.20220106).

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

## A. Related Work

### A.1. Time-Series Forecasting

Time-series forecasting has been studied under several modeling paradigms. Deep learning methods automatically extract temporal representations with neural networks and have achieved strong performance, but their reliance on task-specific training limits their adaptability (Huang et al., 2024; Liu et al., 2023). Large pretrained time-series models, such as TimesFM (Das et al., 2023) and Time-MoE (Shi et al., 2024), are pretrained on large-scale time-series corpora to support zero-shot and few-shot forecasting across domains. However, their performance, especially for multivariate forecasting with complex cross-channel dependencies, can still lag behind specialized models (Yang et al., 2026). A complementary line of work explores in-context forecasting, where models adapt to a forecasting task through the provided context (Zhu et al., 2023; Feuer et al., 2024). TabPFN-TS follows this paradigm by reformulating forecasting as tabular regression, and TabPFN-TSP further incorporates FFT-based periodic features. Nevertheless, these methods still mainly rely on explicit temporal features, which may be insufficient for non-stationary and event-driven patterns and can dilute informative local signals in large contexts.

### A.2. Retrieval-Augmented Models

Retrieval augmentation enhances predictive models by conditioning them on relevant external evidence rather than relying solely on parametric knowledge (Lewis et al., 2020). In time-series forecasting, this paradigm is particularly appealing because analogous historical patterns can provide direct evidence for future dynamics. Recent studies have explored incorporating retrieval mechanisms into time-series forecasting, typically by retrieving relevant historical patterns and adapting forecasting models to downstream tasks through fine-tuning (Liu et al., 2024; Han et al., 2025; Yang et al., 2025). However, existing studies primarily focus on time-series foundation models, leaving its role in TabPFN-TS-style in-context forecasting underexplored. Our work addresses this gap by exploring retrieval augmentation for TabPFN-based forecasting in a fine-tuning-free manner, aiming to mitigate its reliance on explicit temporal features and global context summarization.

## B. More Experiment Results and Analysis

### B.1. More Qualitative Analysis

Beyond the qualitative results in Section 3.2.2, we conduct additional synthetic experiments following TabPFN-TS and RAFT, as shown in Figure 3. These experiments are designed to reduce dataset-specific confounding factors and provide a clearer examination of the characteristics, advantages, and limitations of TabPFN-TSRA under different temporal patterns.

**Linear trend with noise.** Figure 3(a) shows that TabPFN-TSRA captures the overall trend direction better than the two baselines. However, its absolute prediction level is less accurate than TabPFN-TSP in this case, which also exposes a limitation of our method. Under level-shift correction, different historical windows of a near-linear trend become highly similar after being aligned to their last observations, so retrieval tends to select many adjacent windows that provide largely redundant shape information. As a result, the retrieved analogues help TabPFN-TSRA recognize the increasing trend, but provide limited additional evidence for determining the correct future level. The task is therefore effectively transformed into an offset-based extrapolative regression problem, which is challenging for TabPFN because its in-context prediction is better suited to interpolation within the observed target range than to extrapolating beyond the conditioning values. In contrast, TabPFN-TSP constructs only a small number of temporal feature columns, making the small-sample regression problem simpler for estimating the absolute level in this setting, while TabPFN-TS produces nearly flat predictions and fails to reflect the linear trend.

**Exponential trend.** Figure 3(b) shows that TabPFN-TSRA performs best in the exponential-trend setting. Compared with TabPFN-TS and TabPFN-TSP, which both produce nearly flat forecasts and fail to capture the accelerating growth pattern, TabPFN-TSRA better follows the upward trend by retrieving historical analogues with similar local growth shapes and using their subsequent trajectories as evidence for extrapolation. Compared with Figure 2(b) and Figure 3(a), this result suggests that retrieval augmentation is particularly useful when the future trajectory depends on non-purely-linear trend evolution rather than purely linear extrapolation, because similar historical subsequences can provide informative continuation patterns beyond fixed temporal features or global context modeling.

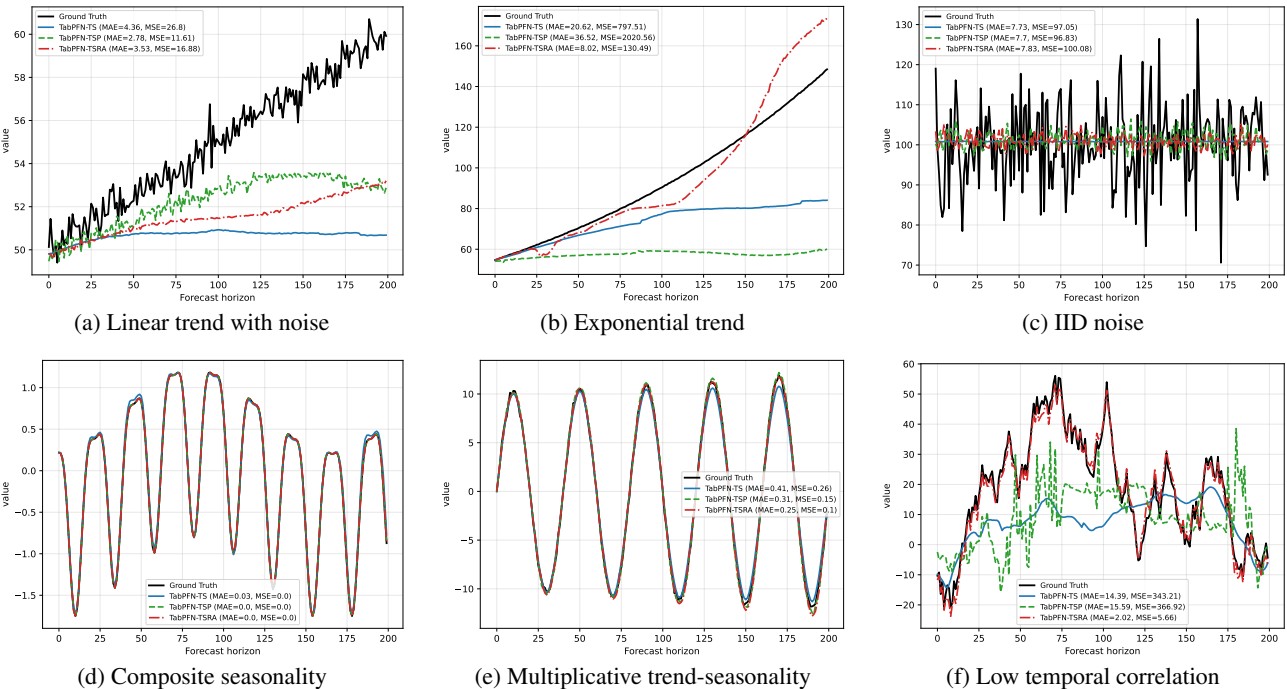

*Figure 3*. Additional qualitative analysis on synthetic time series. These controlled experiments cover representative temporal patterns, including noisy trend, exponential trend, IID noise, composite seasonality, multiplicative trend-seasonality with increasing amplitude, and low temporal correlation.

**IID noise.** Figure 3(c) shows that TabPFN-TSRA performs worst in the pure-noise setting. This is expected because fully independent noise is inherently unsuitable for retrieval augmentation: similar-looking historical subsequences do not imply similar future continuations. Therefore, the retrieved analogues introduce little useful temporal evidence and may even amplify irrelevant variations. However, comparing with Figure 2(b) and Figure 3(a) shows that, in real or non-pure-noise scenarios, TabPFN-TSRA still produces smoother forecasts and is more robust to noise perturbations than TabPFN-TSP.

**Composite seasonality.** Figure 3(d) shows that all three methods perform well on composite seasonal patterns. This indicates that regular periodic structures can be effectively captured by TabPFN-based forecasting models. Among them, TabPFN-TSRA achieves competitive performance with TabPFN-TSP, which explicitly uses FFT-based period features, and slightly outperforms TabPFN-TS. This suggests that retrieval augmentation can preserve useful periodic evidence from historical analogues without relying solely on explicit frequency-based features.

**Multiplicative trend-seasonality.** Figure 3(e) shows that TabPFN-TSRA performs best under multiplicative trend-seasonality. The three methods show comparable ability to capture the seasonal component, but TabPFN-TSRA better follows the increasing trend and the changing amplitude than the two baselines. Compared with the purely linear setting in Figure 3(a), this result suggests that TabPFN-TSRA is more effective when the trend is coupled with non-linear or periodic structures, rather than appearing as a purely linear extrapolation problem. In such cases, the retrieved analogues provide more informative continuation patterns, allowing TabPFN-TSRA to exploit both seasonal regularity and trend evolution.

**Low temporal correlation.** Following RAFT (Han et al., 2025), Figure 3(f) considers a setting with low temporal correlation, where local changes are less smooth and cannot be easily inferred from nearby observations alone. This setting is similar to random-walk-like short-term patterns: the sequence may not exhibit strong deterministic temporal dependence, but similar change patterns can recur at distant historical positions. TabPFN-TSRA performs better in this case because retrieval allows the model to search beyond the immediate context and use historically similar continuations as external evidence. Compared with TabPFN-TS and TabPFN-TSP, which mainly rely on the current context or a limited feature construction, TabPFN-TSRA can exploit these weak but recurring analogues more effectively. This result suggests that retrieval augmentation is especially useful when temporal correlation alone is insufficient, but historical repetitions of similar local patterns still exist.

**Takeaway.**  Overall, these experiments show that TabPFN-TSRA has clear advantages in realistic time-series scenarios where multiple temporal structures coexist. For **trend** forecasting, TabPFN-TSRA provides the strongest ability to follow trend evolution, especially when the trend is mixed with seasonality, exponential growth, or changing amplitude, which is more aligned with real-world time series. For **seasonality**, TabPFN-TSRA achieves performance comparable to TabPFN-TSP, which explicitly uses FFT-based period features, while avoiding a strong dependence on hand-designed frequency features. For **noise**, TabPFN-TSRA is less suitable than TabPFN-TS under pure IID noise, because retrieval cannot provide meaningful analogues when no temporal structure exists; nevertheless, in realistic non-pure-noise settings, it produces smoother and more stable forecasts than TabPFN-TSP. Moreover, TabPFN-TSRA can handle challenging cases such as **non-stationary and event-like patterns** and **low temporal correlation**, where useful historical evidence may exist at distant positions but is difficult to capture through global context construction or explicit periodic features. These results highlight the advantage of TabPFN-TSRA in leveraging local historical evidence for non-stationary, partially recurring, and structurally complex time-series dynamics.

### B.2. Hyperparameter Study

To analyze the sensitivity of TabPFN-TSRA to retrieval-related hyperparameters, we vary the number of retrieved analogues $k$, the minimum similarity threshold $\tau$, and the train length $l$. Figure 4 presents the results on several representative datasets.

We summarize the main observations as follows. **First**, TabPFN-TSRA is relatively insensitive to the number of retrieved analogues $k$. Across most datasets, the normalized MSE remains close to the best value as $k$ varies, suggesting that the retrieval module is stable once sufficiently similar candidates are selected. This indicates that the method does not depend heavily on a specific retrieval size, as long as the retrieved analogues provide relevant historical evidence. **Second**, the minimum similarity threshold $\tau$ has a clearer but still dataset-dependent effect. Moderate thresholds such as 0.8 and 0.9 usually achieve stable performance, while an overly strict threshold such as 0.98 can degrade performance on some datasets, because too few analogues may be retained and useful retrieval diversity may be lost. This effect is especially visible on ETTm1, where the normalized MSE increases as the threshold becomes stricter. **Finally**, the train length $l$ is the most sensitive hyperparameter. Different datasets prefer different temporal scales: too short lengths may fail to capture sufficient temporal structure, whereas too long lengths can introduce irrelevant historical patterns. For example, Weather benefits from a longer train length, while Exchange shows large degradation under some intermediate or long settings. These results indicate that TabPFN-TSRA is robust to $k$, but benefits from selecting $\tau$ and especially $l$ according to the similarity density and temporal scale of each dataset.

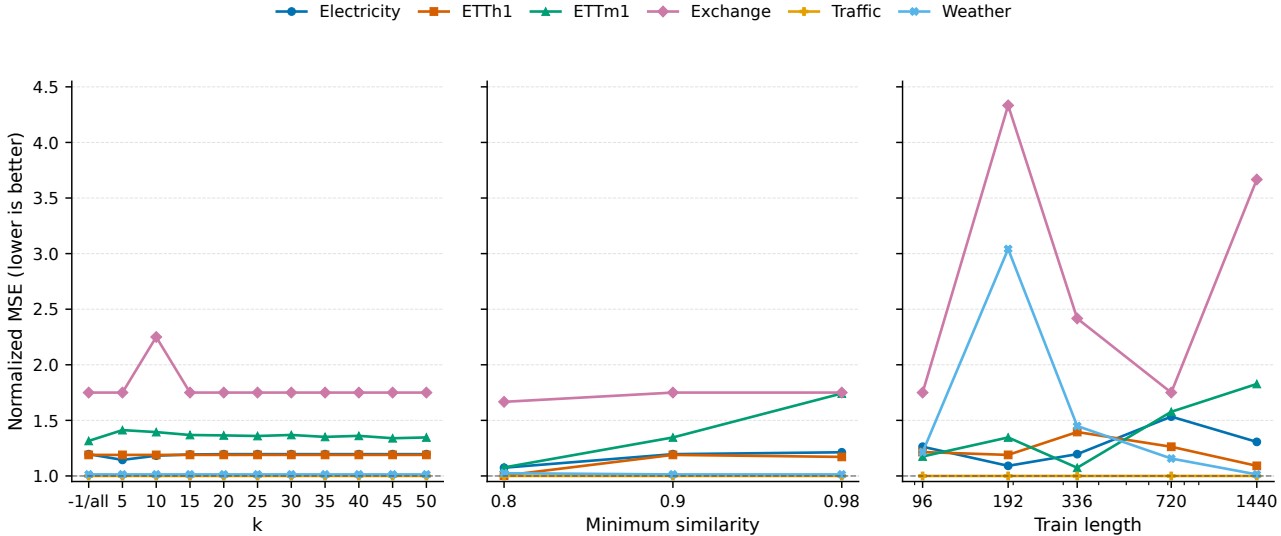

*Figure 4.* Hyperparameter study of TabPFN-TSRA. We analyze the influence of three retrieval-related hyperparameters: the number of retrieved analogues $k$, the minimum similarity threshold $\tau$, and the train length $l$. All MSE values are normalized for better visualization.

Table 2 reports the best performance of TabPFN-TSRA under different choices of retrieval-related hyperparameters, including the number of retrieved sequences $k$, the minimum similarity threshold $\tau$, and the train length $l$. Specifically, we use a fixed

search space across all datasets, where $k \in \{5, 10, \ldots, 50\}$, with an additional setting $k = -1$ that keeps all candidates satisfying the similarity constraint, and $\tau \in \{0.8, 0.9, 0.98\}$. Following TabPFN-TSP and RAFT (Han et al., 2025), we set the train length according to the forecasting horizons, i.e., $l \in F \cup \{2 \times \max(F)\}$. For AirPassengers, we exclude the one-step forecasting horizon from this train-length search space, since a single time step cannot form a meaningful temporal pattern for retrieval. For TabPFN-TSP, we similarly select the best result over the same train-length search space. The results obtained under these best-performing configurations are consistent with our main findings, further confirming the effectiveness of retrieval augmentation across datasets.

*Table 2.* Best forecasting results on 11 benchmark datasets within the predefined hyperparameter search space. We report the average results over all forecasting horizons. Lower MSE and MAE indicate better performance. The best results are highlighted in bold. Scientific notation with e-notation is used for large values.

| Dataset | TabPFN-TSRA | | TabPFN-TSP | | TabPFN-TS | |
|---|---|---|---|---|---|---|
| | MAE | MSE | MAE | MSE | MAE | MSE |
| Traffic | **0.0024** | **0.0000** | 0.0025 | **0.0000** | 0.0032 | **0.0000** |
| Exchange | **0.0321** | **0.0013** | 0.0428 | 0.0027 | 0.0503 | 0.0046 |
| Electricity | 189.00 | 7.13e4 | **175.43** | **5.89e4** | 181.02 | 6.44e4 |
| ETTh1 | **2.22** | **8.43** | 2.29 | 8.71 | 3.21 | 15.76 |
| ETTh2 | **4.83** | **39.59** | 5.12 | 43.70 | 5.04 | 44.34 |
| ETTm1 | **1.58** | **4.60** | 1.88 | 6.25 | 1.72 | 5.28 |
| ETTm2 | **3.56** | **23.86** | 4.96 | 39.76 | 3.81 | 26.51 |
| Stock | **1.443** | **4.110** | 1.868 | 6.202 | 1.805 | 6.130 |
| Illness | **1.37e5** | **3.66e10** | 1.87e5 | 5.45e10 | 3.14e5 | 1.79e11 |
| AirPassengers | **12.85** | **268.30** | 51.13 | 5071.01 | 15.45 | 360.79 |
| Weather | 9.77 | 174.50 | **9.05** | **150.88** | 9.37 | 166.64 |

## B.3. Regressor Replacement Study

To further examine the role of the regression model in TabPFN-TSRA, we conduct a model-replacement study by feeding the same temporally aligned feature matrices under the best hyperparameter configuration of TabPFN-TSRA into several widely used regression models. This setting provides relatively high-quality retrieved analogues and thus allows us to focus on how different regression models exploit the same retrieval-augmented representation. The compared models include CatBoost (Prokhorenkova et al., 2018), Random Forest (Breiman, 2001), LightGBM (Ke et al., 2017), XGBoost (Chen & Guestrin, 2016), linear regression (Weisberg, 2005), SVM (Steinwart & Christmann, 2008), and a mean-value baseline. This experiment aims to determine whether the retrieved analogue representation is generally useful across different regressors, and more importantly, whether the in-context modeling capability of TabPFN-v2 provides additional benefits beyond the retrieval module itself. Table 3 reports the comparison between TabPFN-TSRA and its regression-model variants.

The results show several findings. **First**, TabPFN-TSRA achieves the best overall performance among all regression variants, obtaining the best MAE on 8 out of 11 datasets and the best MSE on 7 out of 11 datasets, which suggests that TabPFN-v2 can effectively exploit the temporally aligned feature matrices constructed from retrieved analogues. **Second**, tree-based models such as CatBoost and Random Forest remain competitive on several datasets, indicating that the retrieved analogue representation itself contains useful predictive information beyond a specific regression model. **Finally**, simpler models such as linear regression, SVM, and the mean baseline are less stable, especially on non-stationary or trend-dominated datasets. In particular, the numerical explosion of linear regression on ETTm1 indicates that directly fitting an unconstrained linear model on highly correlated retrieved analogue features can be unstable. These results suggest that retrieval provides useful local evidence, while the in-context modeling capability of TabPFN-v2 is important for fully leveraging it.

## C. More Implementation Details

**Dataset Settings.** Following TabPFN-TS, we chronologically split each dataset by using the last 10% of the series as the test portion and the first 90% as the initial historical sequence. Forecasting windows are then constructed in a rolling manner, where the available historical sequence is progressively expanded as the window moves forward. This fixed chronological split ensures that no future observations from the test portion are used during input construction. We also follow the same preprocessing protocol as TabPFN-TS, including the handling of missing values.

*Table 3.* Results of TabPFN-TSRA with different regression models on 11 benchmark datasets. We report the average results over all forecasting horizons. Lower MSE and MAE indicate better performance. The best results are highlighted in bold. Scientific notation with e-notation is used for large values.

| Dataset | TabPFN-v2 | | CatBoost | | RandomForest | | LightGBM | | XGBoost | | Linear | | SVM | | Mean | |
|---|---|---|---|---|---|---|---|---|---|---|---|---|---|---|---|---|
| | MAE | MSE | MAE | MSE | MAE | MSE | MAE | MSE | MAE | MSE | MAE | MSE | MAE | MSE | MAE | MSE |
| Traffic | **0.0024** | **0.0000** | 0.0042 | 0.0001 | 0.0041 | 0.0001 | 0.0047 | 0.0001 | 0.0044 | 0.0001 | 0.0044 | 0.0001 | 0.0261 | 0.0011 | 0.0212 | 0.0006 |
| Exchange | **0.0321** | **0.0013** | 0.0362 | 0.0024 | 0.0362 | 0.0023 | 0.0388 | 0.0026 | 0.0363 | 0.0023 | 0.0384 | 0.0027 | 0.0507 | 0.0041 | 0.0543 | 0.0047 |
| Electricity | 189.00 | 7.13e4 | **181.33** | **6.47e4** | 187.10 | 6.98e4 | 188.57 | 7.00e4 | 192.92 | 7.24e4 | 219.67 | 9.35e4 | 327.48 | 1.75e5 | 382.32 | 2.13e5 |
| ETTh1 | **2.22** | **8.43** | 2.27 | 8.62 | 2.26 | 8.60 | 2.27 | 8.79 | 2.30 | 8.85 | 2.37 | 9.63 | 2.34 | 9.06 | 2.31 | 8.83 |
| ETTh2 | 4.83 | 39.59 | 4.88 | **38.94** | 4.92 | 39.38 | 4.93 | 39.37 | 5.02 | 40.91 | 6.18 | 70.62 | 4.98 | 40.69 | 5.79 | 53.03 |
| ETTm1 | 1.58 | 4.60 | 1.56 | 4.50 | **1.54** | **4.37** | 1.55 | 4.44 | 1.58 | 4.51 | 7.97e8 | 9.90e19 | 1.61 | 4.69 | 1.83 | 5.81 |
| ETTm2 | **3.56** | **23.86** | 3.73 | 25.62 | 3.66 | 25.07 | 3.69 | 25.88 | 3.76 | 27.08 | 6.04 | 86.83 | 3.99 | 28.64 | 4.92 | 38.88 |
| Stock | **1.443** | **4.110** | 1.497 | 4.759 | 1.522 | 4.762 | 1.799 | 6.436 | 1.627 | 5.572 | 1.569 | 4.293 | 1.556 | 5.218 | 2.031 | 7.708 |
| Illness | **1.37e5** | **3.66e10** | 1.57e5 | 4.23e10 | 1.56e5 | 4.24e10 | 1.69e5 | 4.62e10 | 1.61e5 | 4.27e10 | 1.57e5 | 4.27e10 | 3.01e5 | 1.12e11 | 2.95e5 | 1.09e11 |
| AirPassengers | **12.85** | **268.30** | 20.58 | 865.04 | 22.39 | 1013.12 | 63.70 | 7895.75 | 28.46 | 1292.81 | 21.99 | 952.13 | 68.61 | 9126.04 | 63.70 | 7895.75 |
| Weather | 9.77 | 174.50 | 9.59 | 168.62 | 9.79 | 174.00 | 9.75 | 173.19 | 9.71 | 172.15 | 19.85 | 8051.42 | **9.54** | **164.87** | 9.96 | 177.44 |

**Train-Length Settings.** For TabPFN-TSRA, the train length $l$ determines the length of both the query subsequence and each retrieved historical analogue, and therefore controls the temporal scale used for retrieval and feature-matrix construction. Following TabPFN-TSP and RAFT (Han et al., 2025), we select $l$ from the horizon-dependent search space $F \cup \{2 \times \max(F)\}$, while also considering the dataset length and its dominant temporal characteristics. In the final configuration, we set $l = 336$ for Electricity and Traffic, $l = 192$ for the four ETT datasets, $l = 96$ for Exchange Rate, $l = 24$ for Stock and AirPassengers, $l = 120$ for Illness, and $l = 1440$ for Weather. These choices reflect different temporal scales across datasets: longer train lengths are used for datasets with longer histories or stronger long-range periodic structures, while shorter train lengths are adopted for relatively short datasets to avoid constructing retrieval candidates from overly long and less reliable contexts. For fairness, TabPFN-TSP uses the same train-length setting.

**TabPFN-TS Settings.** For TabPFN-TS, we use the official implementation released by the authors. Following its original setting, the model is given access to the most recent 4096 observations after removing missing points. When the available historical sequence contains fewer than 4096 observations, we use all available observations. This setting follows the original TabPFN-TS protocol and serves as the vanilla TabPFN-based forecasting baseline without retrieval augmentation.

**TabPFN-TSP Settings.** For TabPFN-TSP, we follow the original implementation, where the train lengths are set according to the forecasting horizons, i.e., $l \in F \cup \{2 \times \max(F)\}$, which is also consistent with the setting used in TabPFN-TSRA. In its original design, TabPFN-TSP selects the top-$k$ frequencies with the highest amplitudes and computes the corresponding periods according to the reciprocal relationship between frequency and period. Following the original setting, we set $k = 1$, which corresponds to selecting the single dominant frequency.

**Fallback Settings.** Due to the characteristics of time-series data, the retrieval-based and frequency-based construction procedures do not always guarantee a valid candidate or period for feature-matrix construction. For TabPFN-TSRA, when no candidate sequence in the retrieval pool satisfies the similarity threshold, we use the top-3 most similar candidates with positive correlations as a fallback; when all candidate correlations are negative, we only keep the single most similar candidate. For TabPFN-TSP, we observe that when $k = 1$, the resulting period can sometimes cover almost the entire sequence, making the construction of the corresponding feature matrix infeasible. In such cases, we set $k = 2$ as a fallback.

**Machine Learning Regressor Settings.** Unless otherwise specified, we use the default hyperparameters of the corresponding libraries. The only additional preprocessing is applied to SVM, where the input features are standardized before regression. For tree-based regressors, including Random Forest, XGBoost, LightGBM, and CatBoost, the feature matrices are used directly, with parallel execution enabled when supported. In CatBoost, we use a learning rate of `0.01`; other unmentioned hyperparameters are kept at their default values.

