# OpenReview forum: "TabPFN-TSRA: Retrieval-Augmented TabPFN for Time-Series Forecasting"
_ICML.cc/2026/Workshop/FMSD — FMSD @ ICML 2026 Poster_

### Official Review · Reviewer_UEXW · 2026-05-20

**Rating:** 6
**Confidence:** 4

**Review:**

### Summary

The main contribution of this paper is proposing a new in-context-learning/RAG formulation for univariate time series forecasting over TabPFN. The paper builds on the existing work of  TabPFN-TS and existing RAG formulations in time series forecasting. The paper uses the most recent observed segment as a query and retrieves historically similar sequences by correlation (with value normalization by subtracting the last observation). Then, they pair those sequences with their subsequent observations, and construct feature matrices for TabPFN to forecast the horizon. At each row, they represent one time step, with the original sequence's value at that time step as the label.

### Strengths

1. I find the simplicity of RAG formulation to be appealing. TabPFN-TSRA has  good performance despite this simple formulation. Despite not being truly novel as RAG has been studied extensively in time-series, a simple yet effective formulation is interesting to the community.

2. The authors conducted hyperparameter studies, and also tried different regressors. I think their finding here that TabPFN is stronger among the variants they tried, followed by gradient boosting methods and that simple linear regression to be unstable to be interesting. I think studying further why TabPFN is effective for this formulation would be interesting.

3. I also find the synthetic study to be valuable. Real datasets, though the ultimate goal, sometimes are less suitable to show the model/formulation's overall behavior. One additional thing authors could try is to outline exactly where their RAG formulation benefits: under what regime it has superior performance over TabPFN-TS or non-RAG variants.

### Areas for Improvement

1. A comparison with Chronos-2 would definitely be required for a non-workshop submission. Chronos-2 natively handles ICL and it would be interesting to show how the existing formulation here work there and also how does TabPFN vs Chronos-2 compare in this setting.
2. The paper subtracts the last value from both the query and candidate segments and then computes pearson correlation. However, isn't pearson correlation already invariant to additive shifts? So subtracting the final value before computing pearson correlation should not change the similarity score.
3. The LTSF benchmark has been long shown to include limited data diversity. Evaluation on GIFT-Eval would help justifying the paper's contributions.
4. The method also appears to be quite sensitive to the train/query length.

### Detailed Comments

The paper would be stronger with ablations for the retrieval methodology such as: no level-shift correction, pearson vs euclidean vs cosine similarity or dynamic-time-warping. Especially dynamic time warping addresses the issue in retrieving sequences with similar characteristics but different in scale etc. There is an extensive work in finding similar motifs in time series analysis (please look at motif discovery and the existing work such as https://arxiv.org/abs/2009.07907).

Also, since the method retrieves historical analogues and their future continuations, a natural baseline is a direct nearest-neighbor forecast, such as averaging the continuations of the retrieved segments after level alignment.

### Justification of Score

The paper is interesting to the workshop community because of its simple RAG formulation for time series forecasting over TabPFN. But, due to the the areas of improvement and detailed comments I outline, I am giving marginally above acceptance score.

---

### Official Review · Reviewer_U68N · 2026-05-22
**A reasonable, training-free retrieval-augmented TabPFN idea with honest failure analysis, but with narrow baselines, an overstated 82% win-rate that hides losses, and missing statistical testing.**

**Rating:** 6
**Confidence:** 3

**Review:**

## Summary of contributions

The paper proposes **TabPFN-TSRA**, a retrieval-augmented, fine-tuning-free framework for TabPFN-based time-series forecasting. Rather than relying on explicit temporal features (calendar variables, FFT-based periods) as in TabPFN-TS/TabPFN-TSP, it retrieves analogous historical subsequences using level-shift-corrected Pearson-correlation matching, then organizes the matched segments and their continuations into temporally aligned feature matrices that serve as in-context regression examples for TabPFN. On 11 univariate benchmarks it reports an 82% MAE win rate over the two TabPFN baselines, supported by a hyperparameter sensitivity study, a regressor-replacement ablation, and qualitative analyses on synthetic patterns.

## Strengths
- **Clean, well-motivated idea.** Combining retrieval augmentation with training-free TabPFN forecasting is natural and underexplored; the level-shift correction plus temporally aligned feature-matrix construction is a sensible mechanism for injecting local evidence without explicit temporal features.
- **Informative control experiment.** The regressor-replacement study (feeding the same retrieved features to CatBoost, Random Forest, LightGBM, XGBoost, linear, SVM, and a mean baseline) isolates the value of TabPFN's in-context learning and is a useful, generalizable result.
- **Honest characterization of limitations.** The paper clearly documents where the method fails (pure IID noise; absolute-level estimation under purely linear trends), which strengthens trust in the analysis.

## Weaknesses
1. **Very narrow baselines.** Only TabPFN-TS and TabPFN-TSP are compared. There are no classical baselines (naive/seasonal-naive/ARIMA/ETS), no time-series foundation models (Chronos/TimesFM/Moirai), and no other retrieval-augmented forecasters (e.g., RAFT, TimeRAG), several of which are cited. This makes the method's absolute quality harder to judge, though the relative gains over both TabPFN baselines remain consistent.
2. **Un-normalized, raw-scale metrics.** Keeping original scales makes MAE/MSE span many orders of magnitude and incomparable to published results on these standard benchmarks; it also makes the aggregate "win rate" sensitive to a few large-magnitude datasets.
3. **The headline "82% win rate" is fragile and undefined.** The denominator is unspecified, the metric is binary (ignoring magnitude/ties and variance), and it masks clear losses, e.g., the method underperforms TabPFN-TS on Electricity and Weather, and TabPFN-TSP on Electricity even under best-config (Table 2).
4. **No variance or significance testing.** Many wins are within plausible run-to-run noise; no seeds, error bars, or tests are reported.

## Suggestions
- Add classical baselines (at least naive/seasonal-naive) and one or two time-series foundation models, plus at least one existing retrieval-augmented forecaster, so absolute performance and the specific benefit of retrieval can be judged.
- Report standard normalized metrics on the conventional benchmark protocol, alongside the raw-scale numbers, to enable comparison with the literature.
- Replace (or supplement) the win-rate headline with mean ± std over multiple seeds and a paired significance test; define the win-rate denominator explicitly and report per-dataset wins/losses.